# Alternating Gemcitabine/Nab-Paclitaxel (GA) and 5-FU/Leucovorin/Irinotecan (FOLFIRI) as First-Line Treatment for De Novo Metastatic Pancreatic Cancer (MPC): Safety and Effect

**DOI:** 10.3390/cancers15235588

**Published:** 2023-11-26

**Authors:** Brett A. Schroeder, Margaret T. Mandelson, Vincent J. Picozzi

**Affiliations:** 1Virginia Mason Medical Center, Seattle, WA 98101, USA; mmandelson@benaroyaresearch.org (M.T.M.); vincent.picozzi@commonspirit.org (V.J.P.); 2National Cancer Institute, National Institutes of Health, Bethesda, MD 20892, USA; 3Benaroya Research Institute, Seattle, WA 98101, USA

**Keywords:** metastatic, pancreatic, cancer, gemcitabine, FOLFIRI

## Abstract

**Simple Summary:**

This study explores the efficacy and safety of alternating gemcitabine/nab-paclitaxel (GA) and 5-FU/leucovorin/irinotecan (FOLFIRI) regimens in patients with metastatic pancreatic cancer (MPC). The approach aims to reduce toxicity, slow resistant cancer biology, and facilitate the incorporation of other therapeutic agents. The results show a median overall survival (mOS) of 13.2 months, competitive with standard regimens, and encouraging long-term survival rates at 18 and 24 months. The toxicity profile is favorable, and supportive growth factors were not needed. This regimen appears promising for MPC and warrants further investigation.

**Abstract:**

Background: Both gemcitabine- and 5-fluorouracil (5-FU)-based chemotherapy regimens have demonstrated efficacy in metastatic pancreatic cancer (MPC). Alternating these regimens may reduce toxicity, slow resistant cancer biology emergence, and provide a platform for the addition of other therapeutic agents. Alternating gemcitabine/nab-paclitaxel (GA) and 5-FU/leucovorin/irinotecan (FOLFIRI) in MPC has previously been reported at our own institution and elsewhere. An extension of our institutional observations is reported here. Methods: Patient eligibility required the following: biopsy-proven de novo MPC, no prior evidence of disease on CT, ECOG performance status (PS) ≤ 2, and bi-dimensionally measurable disease. Treatment (Tx) entailed gemcitabine 1000 mg/m^2^ and nab-paclitaxel 125 mg/m^2^ 1, (8), 15 alternating every 8 weeks (2 cycles) with FOLFIRI using standard dosing. Patients were radiographically re-staged every 8 weeks. Tx spanned up to 12 cycles. Tx thereafter was decided following patient/physician discussion. Results: Median overall survival (mOS) was 13.2 months (95% CI 10.9–16.5 months). Median progression-free survival (mPFS) was 8.5 months (95% CI, 7.1–9.9). The 6-, 12-, 18-, and 24-month OS rates were 88%, 54%, 36%, and 20%, respectively. The disease control rate at 16 weeks was 83% (37% PR, 46% SD). Hematologic toxicity grade ≥ 3 included 9.3% anemia, 10.2% neutropenia, and 4.6% thrombocytopenia. Neutrophil growth factors were not used in this cohort. Non-hematologic toxicities grade ≥ 3 included neuropathy 0.9%, nausea/vomiting 0.9%, and diarrhea 0.9%. No patients experienced mucositis on this regimen. Conclusions: Alternating GA/FOLFIRI in MPC has a favorable toxicity profile in comparison to current standard regimens. Median OS was at least competitive with standard regimens, and longer-term (18 and 24 months) OS seemed particularly encouraging. Treatment for ≥48 weeks and ECOG PS of zero at the time of treatment initiation were prognostically significant. Further investigation using this regimen including randomized comparisons, the incorporation of molecular data, and use of additional agents is merited.

## 1. Introduction

Pancreatic adenocarcinoma (PDAC) represents a significant global health challenge, affecting more than 450,000 individuals worldwide, with an impact that continues to grow [1]. In 2019, it emerged as the third leading cause of cancer-related mortality in the United States, and was projected to be the second leading cause of cancer death by 2020 [2]. This underscores an urgent need for more effective treatments. PDAC is notorious for its ruthless progression, characterized by an alarmingly short median overall survival (mOS) of merely 6 months [3]. For patients with metastatic disease, the prospects are even more dismal, with 1-year and 5-year survival rates hovering around a meager 25% and 2%, respectively [4]. Although some regions, such as the United States and Canada, have witnessed slight increases in overall survival, these trends have not been universally observed [5]. The therapeutic landscape for metastatic pancreatic adenocarcinoma has, regrettably, evolved at a glacial pace, marked by modest, incremental improvements in patient outcomes over the past decade.

The last decade has witnessed modest improvements in OS for metastatic pancreatic cancer (MPC), largely due to the development of more active chemotherapy combinations [6,7,8,9,10]. This protracted struggle against PDAC can be attributed, in part, to an array of biological and clinical challenges that have confounded researchers and clinicians alike. With respect to biological factors, the genetic heterogeneity observed in PDAC is a formidable hurdle. The disease is genetically complex, with great biologic heterogeneity. Key driver mutations such as KRAS are just now becoming amenable to therapeutic targeting [11]. The tumor microenvironment in pancreatic cancer is particularly intricate, featuring a dense desmoplastic stroma that impedes drug penetration and fosters an immunosuppressive milieu. It is immunologically “cold” and metastasizes at an early point in time, limiting the role of local therapies such as surgery and radiation therapy. Finally, the disease has a complex systemic biology, with an often-immense systemic impact on patients, affecting both quality and quantity of life. Furthermore, the absence of reliable biomarkers for early detection and prognosis compounds the problem, often leading to delayed interventions that occur at an advanced, less-treatable stage of the disease. However, a recent study utilizing genomics-driven precision medicine shows that prospective profiling may be possible [12].

Clinical factors further exacerbate the therapeutic challenges associated with metastatic pancreatic adenocarcinoma. PDAC patients are often older, with an average age over 70 years old, and typically have extensive comorbidities and suboptimal treatment tolerances. Pancreatic adenocarcinoma is notorious for its insidious nature, often remaining asymptomatic in its early stages. Consequently, diagnosis is frequently delayed, limiting therapeutic options. Supportive care for PDAC remains challenging, especially for symptoms such as pain and weight loss, and typically requires sophisticated multidisciplinary medical teams which can be challenging to coordinate. Finally, a pervasive sense of nihilism exists surrounding PDAC among medical professionals, the general public, and patients themselves, even to this day.

Efforts to enhance outcomes for patients with metastatic PDAC have evolved incrementally. The lack of consensus on a universally accepted first-line therapy further complicates matters, with treatment decisions influenced by factors such as patient preferences, geographic access to specialized centers, and the availability of clinical trials. Chemotherapy remains the cornerstone of treatment for metastatic PDAC, with combinations of agents such as gemcitabine/nab-paclitaxel and FOLFIRINOX being commonly employed. Recently, in the first positive clinical trial in metastatic PDAC in over a decade, the nal-IRIFox regimen was shown to be modestly superior to gemcitabine/nab-paclitaxel [13]. However, these aggressive combinations are tempered by increased toxicity profiles, rendering them unsuitable for some patients, particularly those with comorbidities or reduced performance status.

Given the above, there is an urgent need to produce superior treatment regimens in PDAC that are (1) able to control the diverse biology of PDAC, (2) are readily tolerated, even by patients of advanced age and significant comorbidity, and (3) have the flexibility to be combined with novel systemic agents. One approach is the use of alternating chemotherapy regimens, a concept originally introduced by Goldie and Coldman [14,15]. This approach offers these characteristics in the treatment of metastatic PDAC. By alternating treatment agents, this approach may mitigate cumulative toxicity, as patients do not receive the same drug for an extended period of time. Additionally, this strategy holds promise in circumventing chemotherapeutic resistance by proactively adapting to changing tumor dynamics. Furthermore, it provides flexibility in incorporating investigational agents with differing mechanisms of action, potentially enhancing treatment efficacy.

Clinical evidence in metastatic PDAC has supported these theoretical considerations as initial investigations into alternating chemotherapy regimens have yielded promising results. Alternating GA and FOLFIRI regimens in both cooperative groups and single institutions [16,17] showed a favorable toxicity profile compared to standard regimens alone, with a 1-year OS of 61%, and a median OS of 16.3 months. Furthermore, researchers showed that alternating FOLFIRINOX with GA had an acceptable toxicity with a complete response in 3.5% of patients and an OS of 17.8 months (95% CI: 11.7–21.3), with a median follow-up of 18.6 months (95% CI: 14.5–25.6) [18].

Given a desire to avoid neurotoxicity, often a limiting factor in the use of platinols in metastatic PDAC, we have focused on the use of gemcitabine/nab-paclitaxel and FOLFRI. In this manuscript, we extend our institutional observations using this approach and provide insights into its potential clinical relevance.

## 2. Methods

This study represents a retrospective review of a single-institution experience. Patients were identified prospectively to receive the regimen under study. Patient eligibility included (1) biopsy-proven de novo MPC without prior evidence of MPC on computed tomography (CT), (2) bi-dimensionally measurable disease, and (3) an Eastern Cooperative Oncology Group (ECOG) [19] performance status of <2.

The regimen studied consisted of the following: gemcitabine 1000 mg/m^2^ and nab-paclitaxel 125 mg/m^2^ 1, (8), 15, alternating every 8 weeks (2 cycles) with the FOLFIRI regimen of irinotecan 180 mg/m^2^, leucovorin 400 mg/m^2^, 5-FU bolus at 400 mg/m^2^, and a continuous 46 h infusion of 2400 mg/m^2^ every 2 weeks. Patients were radiographically re-staged every 8 weeks. Treatment (Tx) was planned for 12 cycles. Tx thereafter was decided per patient and physician preference.

Tumor response was evaluated in patients every 8 weeks by CT, and measurements of carbohydrate antigen 19-9 (CA19-9) [20] level were performed at baseline and every 8 weeks thereafter. Labs were reviewed weekly prior to treatment for treatment-related adverse and serious adverse events based on the National Cancer Institute Common Terminology Criteria for Adverse Events, version 5.0. Laboratory testing was performed weekly, and dose modifications or dose delays were based upon these labs.

Demographic data, performance status, and radiographic treatment response at 8 and 16 weeks were collected from the medical record. Overall survival and PFS were assessed from the date of first treatment to the time of death or last follow-up with Kaplan–Meier curves using STATA 17 [21]. Hazard ratios with corresponding 95% confidence intervals (CIs) for OS and PFS were calculated. The relation between selected characteristics and OS was investigated using Cox proportional hazards. Data collection was closed on 28 February 2023.

## 3. Results

Between October 2015 and October 2020, 108 patients were identified for study inclusion using the above criteria. Patient characteristics (Table 1) included a median age of 68 (range 35–81). ECOG PS 0/≥1 values were 54% and 46%, respectively. The number of disease sites of 1/≥1 in patients was 62% and 38%, with 79% of patients having liver involvement, and 40% of patients with biliary obstruction at the time of presentation. The median CA19-9 was 4598, and 56% of patients had a CA19-9 level greater than 59 times the normal limit.

Toxicity for the regimen is shown (Table 2). Hematologic toxicity grade ≥ 3 included anemia (9.3%), neutropenia (10.2%), and thrombocytopenia (4.6%). Neutrophil growth factors were not used in this cohort. Non-hematologic grade ≥ 3 included neuropathy (1%), nausea/vomiting (1%), and diarrhea (1%).

The median time on treatment was 42 weeks, and 42/108 patients (39%) completed 12 cycles of intended initial therapy, while 26 (24%) continued with treatment thereafter. Sixty three percent of patients who completed 12 cycles of treatment resumed treatment after disease progression.

With a median follow-up of 44.6 months, the median overall survival was 13·2 months (95% CI, 10.9–16.5) (Figure 1). The 6-, 12-, 18-, and 24-month OS rates were 88%, 54%, 36%, and 20%, respectively. The median PFS was 8.5 months (95% CI 7.1–9.9).

The ability to receive all 12 cycles of intended therapy was strongly associated with mOS, 22.2 vs. 9.7 months (logrank = 84.7, *p* < 0.0001). Age, ECOG score, number of metastatic sites, presence of liver metastasis, and biliary obstruction were not associated with OS, independently of treatment completion.

## 4. Discussion

The FOLFIRINOX and gemcitabine/nab-paclitaxel chemotherapy regimens set a worldwide standard for the initial treatment of MPC. The PRODIGE 4/ACCORD 11 trial utilizing FOLFIRINOX therapy demonstrated an OS of 11.1 months versus 6.8 months for gemcitabine monotherapy (HR, 0.57 [95% CI, 0.45–0.73]; *p* < 0.001 [9]. Subsequently, the MPACT trial using the combination of gemcitabine with nab-paclitaxel demonstrated an OS of 8.7 months versus 6.6 months with gemcitabine monotherapy (HR, 0.72 [95% CI, 0.62–0.83]; *p* < 0.001) [10]. Very recently, the NAL-IRIFOX regimen has been demonstrated to be a third standard regimen for metastatic pancreatic cancer [22]. Based on these results, any of these regimens can be considered the first-line treatment for patients with MPC.

Multiple investigators have integrated the drugs in these two regimens in different ways. For example in a phase I study of patients with newly diagnosed advanced pancreatic adenocarcinoma, Safran et al. combined 5-FU, leucovorin, oxaliplatin, and three dose levels of nab-paclitaxel (FOLFOX-A) [23]. They demonstrated minimal grade III neurotoxicity, where 21 of 35 patients (60%) had a partial response, and an OS of 15 months. In a phase I/II trial of 12 treatment-naïve MPC patients, Sahai et al., combined 5-fluorouracil, nab-paclitaxel, leucovorin, oxaliplatin, and bevacizumab (FABLOx) [24]. The investigators found that FABLOx had an adverse event profile consistent with several prior studies [9,10,25], a median PFS of 5.6 months (95% CI, 1.7–11.3), an OS of 9.9 months (95% CI, 4.4–13.2), and a 1-year OS of 38.9% (95% CI, 12.6–65.0). Furthermore, in a study evaluating genotype-guided therapy for previously untreated patients with GI malignancies, Joshi et al. combined 5-FU, leucovorin, irinotecan, and nab-paclitaxel (FOLFIRABRAX) [26]. They demonstrated dose-limiting toxicity in five of twenty-three (22%) patients, with an overall response rate of 31% for all patients. Of the 29 patients with pancreatic cancer, 1 (3%) achieved CR, 9 (31%) had PR, 13 (45%) had SD, and disease control was evident in 23 (79%).

Similarly, to minimize both toxicity and the development of chemotherapy resistance, several trials have alternated 5-FU based therapy with gemcitabine. The phase I/II GABRINOX study investigated the sequential monthly administration of nab-paclitaxel and gemcitabine, followed by FOLFIRINOX [18,27]. While the median OS was promising at 17.8 months, the frequencies of grade 3/4 neutropenia, thrombocytopenia, and diarrhea were higher than anticipated. In the phase II SEENA-1 study, nab-paclitaxel plus gemcitabine was given and alternated with modified FOLFIRI (5-FU, leucovorin, and irinotecan, or nab-paclitaxel plus gemcitabine) followed by modified FOLFIRINOX (no 5-FU bolus), for up to 48 weeks [16]. The median OS was 12.3 months and the safety profile was generally similar to PRODIGE 4/ACCORD 11 and MPACT trials, with common grade ≥ 3 toxicities including neutropenia (43%), anemia (21%), and thrombocytopenia (15%). Most importantly, the recently reported SEQUENCE trial reported from Spain [7,28] showed a statistically significant superiority of gemcitabine/nab-paclitaxel alternating with FOLFOX versus gemcitabine/nab-paclitaxel alone (median OS 13.2 vs. 9.7 months, *p* = 0.023) in 157 randomized patients at the expense of increased ≥ grade 3 hematologic toxicity (neutropenia 46 vs. 24% *p* = 0.004, thrombocytopenia 24 vs. 8% *p* = 0.007).

The approach described here seems to offer multiple advantages over the foundational regimens and their derivatives. First, the toxicity profile was superior to standard regimens, especially with respect to neurologic and hematologic toxicity. Growth factor usage was not necessary, and 36% of patients were able to tolerate therapy for 48 weeks or more. The overall toxicity profile suggests that this regimen could likely be adapted to ECOG PS 2 patients, who in general have been restricted to single or double chemotherapy regimens [29].

Second, survival results were encouraging. Although a median OS of 13.2 months (95% CCI, 10.9–16.5) was lower than in our initial report, it remained competitive along with a progression-free survival of 8.9 months with those achieved from standard regimens. Of particular note are the promising “ends” of the overall survival curve (Figure 1, Table 3). The survival rate at 6 months for this regimen was 87.7%, which is superior in comparison to routinely used regimens [5,6], as is the 19.8% 24-month overall survival. The absence of disease-determined prognostic factors [30] also suggests broad applicability to the MPC patient population.

Multiple precautionary comments are important for the interpretation of the data presented here. This study is limited in design, as a single-arm, single-institution endeavor. Additionally, this relatively small cohort may not be reflective of populations elsewhere. Patients that completed treatment may be self-selective with less than the typical number of comorbidities.

Despite these precautionary comments, the chemotherapy regimen described seems competitive with respect to toxicity, efficacy, and applicability with any regimens employed so far for metastatic PDAC. Alternating GA/FOLFIRI in MPC has a more favorable toxicity profile than any current standard regimen. Median OS with this alternating regimen appears superior to GA alone, and competitive with FOLFIRINOX, with long-term (18 and 24 months) OS that seems particularly encouraging. Both treatment for 48 weeks and ECOG PS of 0 at the time of treatment initiation were prognostically significant. Further investigation using this regimen including randomized comparisons, the incorporation of molecular data, and the addition of additional agents is merited.

## 5. Conclusions

In the absence of new systemic therapies, molecular prediction of therapeutic response [31,32] along with the unique sequencing of extant drugs is an important strategy for therapeutic advancement. In the pursuit of improved therapeutic options for advanced pancreatic cancer, where innovative approaches are necessitated by the scarcity of effective systemic therapies, the integration of molecular prediction of therapeutic response and the strategic sequencing of existing drugs have emerged as promising therapeutic advancements. This study provides compelling evidence in support of alternating chemotherapy regimens in the management of advanced pancreatic cancer. Specifically, the alternating GA/FOLFIRI regimen, evaluated within our single-institution experience, demonstrates remarkable competitiveness across multiple crucial dimensions when compared to well-established standard regimens such as GA, FOLFIRINOX, and NAL-IRIFOX. Moreover, this regimen provides a robust foundation for further exploration through the integration of investigational agents, thereby heralding new prospects for therapeutic innovation.

This regimen has a favorable toxicity profile which stands out as a paramount advantage. The reduced burden of adverse effects has the potential to substantially enhance the quality of life of patients undergoing treatment, a critical attribute in the clinical decision-making process. In addition to tolerability, the regimen delivers competitive response rates, a pivotal indicator of its therapeutic efficacy. The heightened likelihood of achieving a positive response substantiates its candidacy as a viable treatment option, particularly in cases where more aggressive regimens like FOLFIRINOX may be contraindicated due to patient-specific factors.

Among the most salient findings of our study is the positive impact on progression-free and overall survival. The observed outcomes favorably compare with established standard treatments, suggesting that the alternating GA/FOLFIRI regimen may offer a promising alternative. Furthermore, the ease of administration stands as a notable advantage. The simplification of treatment regimens holds the potential to bolster compliance while concurrently alleviating some burdens placed upon healthcare systems. This consideration coincides with the overarching goal of improving the patient experience.

The versatility of this regimen as a platform for the integration of investigational agents also merits special attention, particularly in an era characterized by rapid advancements in cancer research. The ability to seamlessly incorporate novel therapeutics into established regimens represents a pivotal asset. This adaptability not only fosters further innovation but also empowers clinicians to explore novel combinations and targeted therapies that may amplify its therapeutic efficacy.

In summary, the findings derived from this single-institution experience underscore the considerable potential of the alternating GA/FOLFIRI regimen as a valuable addition to the armamentarium of treatments available for advanced pancreatic cancer. Its favorable toxicity profile, impressive response rates, and adaptability to investigational agents collectively position it as a compelling option for deeper investigation and potential integration into clinical practice. While the validation of our findings through more extensive multicenter trials is imperative, our study lays a solid foundation for the advancement of the therapeutic landscape for advanced pancreatic cancer.

## Figures and Tables

**Figure 1 cancers-15-05588-f001:**
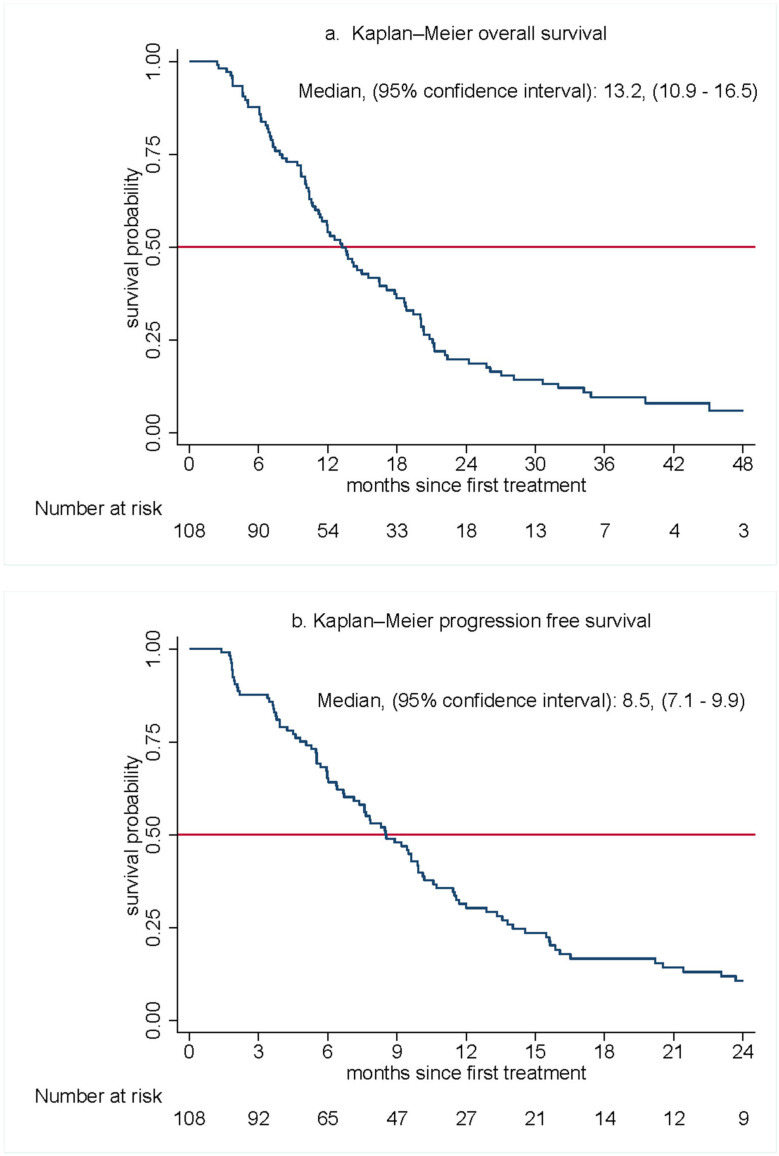
Kaplan–Meier estimates of overall survival (**a**) and progression-free survival (**b**).

**Table 1 cancers-15-05588-t001:** Patient characteristics at baseline.

Demographic and Patient Baseline Characteristics	
Characteristic	(N = 108)
Age—yr	
Median	68
Range	35–81
Sex—no. (%)	
Male	42 (39)
Female	66 (61)
Alive—no. (%)	5 (5)
Dead—no. (%)	91 (84)
Transferred care	12 (6)
* ECOG performance status score—no. (%)	
0	58 (54)
1	49 (45)
2	1 (1)
Site of metastatic disease—no. (%)	
Liver	85 (79)
Lung	14 (13)
Peritoneum	48 (44)
Bone	6 (6)
Number of metastatic sites	
1	67 (62)
2	33 (31)
≥3	8 (7)
Level of carbohydrate 19-9—U/mL ^1^	
Median (95% CI)	4598 (1903–9500)
Range	5–132,000
Biliary obstruction—no. (%)	
yes	43 (40)
no	65 (60)
Thrombosis—no. (%)	
yes	23 (21)
no	85 (79)
Pain at presentation	
yes	78 (72)
no	30 (28)

* ECOG denotes Eastern Cooperative Oncology Group. ^1^ Excludes 3 patients with no CA19-9 expression.

**Table 2 cancers-15-05588-t002:** Safety.

Common Adverse Events of Grade 3 or Higher	
Event	(N = 108)
Adverse event leading to death—no. (%)	0
Grade ≥ 3 hematologic adverse event—no. (%)	
Neutropenia	11 (10.2)
Thrombocytopenia	5 (4.6)
Anemia	10 (9.3)
Febrile neutropenia—no. (%)	2 (1.9)
Thromboembolism	8 (7.4)
Peripheral neuropathy	1 (0.9)
Nausea or vomiting	1 (0.9)
Diarrhea	1 (0.9)
Mucositis	0
Infection	19 (17.6)

**Table 3 cancers-15-05588-t003:** Patient overall survival, progression-free survival, and response rates.

Overall Survival, Progression-Free Survival, and Response Rates	
	All (n = 108)
**Overall Survival (months)**	
Median overall survival—mo (95% CI)	13.2 (10.9–16.5)
6 mo	87.7 (79.8–92.7)
12 mo	54.0 (43.8–63.1)
18 mo	36.2 (26.8–45.7)
24 mo	19.8 (12.4–28.3)
**Progression-Free Survival**	
Median progression-free survival—mo (95% CI)	8.5 (7.1–9.9)
**Progression-Free Survival rate—(95% CI)**	
6 mo	65.2 (55.1–73.6)
12 mo	30.3 (21.6–39.5)
**16-week response ***	
Partial response	39 (37.1)
Stable disease	48 (45.7)
Progressive disease	18 (17.1)
**Disease Control Rate at 16 weeks ***	
No. (%)	87 (82.9)
95% CI	72–87
**Treatment Characteristics**	
Completed 12 cycles of treatment—No. (%)	42 (38.9)
Median OS (95% CI) for completed 12 cycles	22.2 (20.0–28.1)
Did not complete 12 cycles of treatment	66 (61.1)
Median OS (95% CI) for fewer than 12 cycles	9.7 (7.2–10.4)
**Median Follow-Up—mo**	20

* Subjects with missing data excluded from column totals.

## Data Availability

Data are available on reasonable request. All data relevant to the study are included in the article.

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
