# Peer review of "Alternating Gemcitabine/Nab-Paclitaxel (GA) and 5-FU/Leucovorin/Irinotecan (FOLFIRI) as First-Line Treatment for De Novo Metastatic Pancreatic Cancer (MPC): Safety and Effect"

_cancers, 2023, doi:10.3390/cancers15235588_

Round 1
Reviewer 1 Report
Comments and Suggestions for Authors
- line 54: please specify in more detail which factors influence treatment choice (travel?)
- line 80: please specify in more detail the schedule of gem-nab (gemcitabine 1000mg/m2 and nab- 79 paclitaxel 125mg/m2 1, (8), 15). What do you mean with (8)?
- line 116: please provide more info about treatment after progression
Comments on the Quality of English Languagegood
Author Response
Comments and Suggestions for Authors from Reviewer 1
- line 54: please specify in more detail which factors influence treatment choice (travel?)
- Thank you very much, accordingly we have changed “travel” to “proximity to treatment facilities” to help clarify factors that influenced treatment choice.
- line 80: please specify in more detail the schedule of gem-nab (gemcitabine 1000mg/m2 and nab- 79 paclitaxel 125mg/m2 1, (8), 15). What do you mean with (8)?
- We deleted the brackets to alleviate confusion. Of note, gemcitabine/nab- paclitaxel was administered in standard fashion as per package insert.
- line 116: please provide more info about treatment after progression
- The incremental survival can be deduced by subtracting the PFS from the OS. Our apologies, beyond this, are you asking for what fraction of patients received additional therapy after progression on the alternating regimen?
Reviewer 2 Report
Comments and Suggestions for Authors
Well written study with clear rationale.
I have only few concerns.
It is not clear to me what the overlap with previously reported results from the SEENA-1 study is?
Also, from previously published abstracts it appears that control groups were included in the study that received only GA or FFX. Is that correct? The survival data from those groups would rather valuable additions to the current manuscript.
If those control data are not available, a propensity-score matched control group (rather than mentioning the mOS from the FFX and GA studies) would help to interpret the results.
Author Response
- It is not clear to me what the overlap with previously reported results from the SEENA-1 study is?
- Good question, but there is no overlap with this report and the SEENA-1 study, they are separate.
- Also, from previously published abstracts it appears that control groups were included in the study that received only GA or FFX. Is that correct? The survival data from those groups would rather valuable additions to the current manuscript.
- Thank you, but I believe that you might be mistaken. No patients in this report received FFX, nor did any prior publication include control groups (either GA or FFX)
- If those control data are not available, a propensity-score matched control group (rather than mentioning the mOS from the FFX and GA studies) would help to interpret the results.
- Thank you for the feedback, however a propensity score analysis is not possible or relevant in a descriptive study that does not compare two or more groups of patients. Here, we present only 1 group of patients. We did not compare OS or treatment tolerance between groups treated under different regimens.
Round 2
Reviewer 1 Report
Comments and Suggestions for Authors
none
Comments on the Quality of English Languagenone
Author Response
Thank you.
Reviewer 2 Report
Comments and Suggestions for Authors
The uthors could have at least incorporated some text to explain any possible misunderstandings I had in the revised manuscript. In the revised manuscript I only see text deleted?
Author Response
A new file have been uploaded.
Round 3
Reviewer 2 Report
Comments and Suggestions for Authors
I'm still confused by how the SEENA-1 trial compares to what is in the submitted paper?
In the response, the authors claim there is no overlap, yet Table 1 and 2 clearly mention "SEENA-1 in them??Author Response
We agree, SEENA-1 in the tables is confusing and they have been removed. For reference, SEENA-1 was first described in 2017 in JCO, and treated patients initially with gemcitabine/nab-paclitaxel followed by mFOLFIRINOX or alternating mFOLFIRI. For simplicity sake, we have removed reference of this current manuscript to the SEENA-1 study.
Round 4
Reviewer 2 Report
Comments and Suggestions for Authors
No further comments